

**Measurement report:Molecular characterization of organic aerosol**
**in coastal environments using offline FIGAERO-I-CIMS**
Yuping Chen[1,2,3], Lingling Xu[1,2*], Xiaolong Fan[1,2], Ziyi Lin[1,2,3], Chen Yang[1,2,3], Gaojie Chen[1,2,3],
Ronghua Zheng[1,2], Youwei Hong[1,2], Mengren Li[1,2], Yanru Zhang[4], Jinsheng Chen[1,2*]
[1]State Key Laboratory of Advanced Environmental Technology, Institute of Urban Environment,
Chinese Academy of Sciences, Xiamen 361021, China.
[2]Fujian Key Laboratory of Atmospheric Ozone Pollution Prevention, Institute of Urban
Environment, Chinese Academy of Sciences, Xiamen 361021, China.
[3]University of Chinese Academy of Sciences, Beijing 100049, China
[4]Xiamen Environmental Monitoring Station, Xiamen 361021, China
＊Corresponding author.
*E-mail address:* Jinsheng Chen (jschen@iue.ac.cn) and Lingling Xu (linglingxu@iue.ac.cn).





**Abstract.** Organic aerosol (OA), as a key component of particulate matter, exerts significant impacts on public health and the environment. However, understanding of molecular characterization of OA under diverse environments remains limited. This study employed offline FIGAERO-I-CIMS (Filter Inlet for Gases and Aerosols coupled with iodide-adduct Chemical Ionization Mass Spectrometry) to analyze the molecular composition of OA in $PM_{2.5}$ samples collected from a coastal city (urban and seaside sites) in Southeast China during spring 2024. A total of 737 and 768 CHOX compounds were identified at the urban and seaside sites, respectively. CHO compounds dominated in signal intensity (>70%) at both sites, while CHON were more abundant at the urban site and S-containing compounds at the seaside site. The weighted effective oxygen numbers (urban 0.82, seaside 0.85) indicated higher oxidation levels in coastal compounds. Seaside CHOX exhibited lower unsaturation, reduced aromaticity, and higher oxidation states. Categorization showed that urban OA was more influenced by aromatic compounds, whereas seaside OA contained higher proportions of aged aliphatic compounds. Two distinct pollution episodes were selected to investigate CHOX evolution. Case 1 (local accumulation) showed enhanced CHON signals through $NO_3 \cdot$-initiated nighttime oxidation that promoted Aliphatic Nitrates formation, whereas Case 2 (marine air masses) showed increased proportions of Aliphatic-like O-rich CHOX compounds (28% to 39%) via aqueous-phase processing probably under high humidity. These findings advance our understanding of OA molecular characteristics and chemical evolution under different environmental conditions.



**1 Introduction**

The chemical compositions of organic aerosol (OA) are highly complex, exerting distinct impacts on human health and the environment. Parts of OA are emitted directly from natural and anthropogenic sources, known as primary organic aerosols (POA). Additionally, gaseous precursors such as $SO_2$, NO$x$, and volatile organic compounds (VOCs) could be absorbed into atmospheric aerosol and undergo a series of chemical reactions to form secondary organic aerosol (SOA) (Putman et al., 2012; Qi et al., 2017; Xu et al., 2020). Current research has focused more on the quantification and the characteristic of bulk OA (Chazeau et al., 2021; Huang et al., 2014; Sun et al., 2018; Zhou et al., 2020). At the molecular level, OA remains not well understood due to its complex composition, consisting of numerous individual compounds with diverse volatility, functionality, and solubility, and its ultralow atmospheric concentration, which introduces large uncertainties in detection and compound-specific identification (Stark et al., 2017; Xu et al., 2017a; Yu et al., 2016; Zheng et al., 2021). Both anthropogenic and natural sources of OA exacerbate the challenge of their identification and quantification in molecular composition (Daellenbach et al., 2024). Several studies have characterized OA in different environments and have found significant variations in OA molecular composition (Chen et al. 2020; Siegel et al. 2021; Zhang et al. 2024). For example, CHO and CHON compounds dominated urban OA, while S-containing compounds were more abundant in marine aerosols (Siegel et al. 2021; Xin et al. 2024). These studies illustrate that aerosol sources and atmospheric processes can significantly affect the molecular composition of OA. Therefore, accurate OA molecular composition and characterization analyses are essential for advancing the understanding of OA formation mechanisms and providing critical insights into aerosol control strategies (Redman et al., 2002; Wan et al., 2020).

Hard ionization techniques, such as Aerosol Mass Spectrometer (AMS) and Aerosol Chemical Speciation Monitor (ACSM), which are commonly used for online observation of aerosol organic components, cannot provide molecular information of



individual compounds. In contrast, soft ionization techniques overcome this limitation
by enabling observation of OA molecular compounds. The traditional methods such
as Electrospray Ionization-Fourier Transform Ion Cyclotron Resonance Mass
Spectrometry (ESI-FT-ICR MS) and Extractive Electrospray Ionization Mass
Spectrometry (EESI-MS) have been successfully used to characterize OA
compositions due to their ultrahigh mass accuracy and resolution (An et al., 2019; Cui
et al., 2024; Gallimore et al., 2017; Jiang et al., 2016; Ning et al., 2023). With the
development of instruments, ultra-high resolution mass spectrometry has been
increasingly adopted in the field observation of OA at the molecular level
(Daellenbach et al., 2024; Xin et al., 2024; Ye et al., 2021; Zheng et al., 2021). Among
these, Filter Inlet for Gases and Aerosols-Chemical Ionization Mass Spectrometry
equipped with reagent ion iodide (FIGAERO-I-CIMS) has emerged as a promising
technology to detect organic compounds with high acidity or polarity in OA in recent
years (Lee et al. 2014; Lopez-Hilfiker et al. 2014; Bianchi et al. 2019; Du et al., 2022).
FIGAERO-I-CIMS performs direct thermal desorption of filter samples, which
reduces potential sample or compositional losses associated with conventional
pretreatment procedures. Compared to ESI-FT-ICR MS, the OA compounds detected
by FIGAERO-I-CIMS were more oxidized and saturated (Xin et al., 2024).
FIGAERO-I-CIMS can operate in both online and offline modes. At present, the
longest known online operation was a seven-month observation conducted by
Daellenbach et al. (2024) in Beijing. However, this prolonged operation faced some
deficiencies, as the stability and airtightness of the instrument deteriorated over time.
Compared to online mode, the offline mode of FIGAERO-I-CIMS lowers operating
and maintenance costs and provides greater convenience for detecting samples from
different environments within a short period of time. Recent studies have reported the
employments of FIGAERO-I-CIMS in offline mode, e.g., at an urban background site
during summer and winter in Stuttgart City, Germany (Huang et al. 2019), at an urban
site in Beijing, China, under varying pollution levels (Cai et al. 2022), and on the
route near the North Pole (Siegel et al. 2021). To date, research on the molecular



composition of OA under varying environmental conditions remains quite limited.
In this study, offline FIGAERO-I-CIMS was applied to characterize OA at the
molecular level in PM$_{2.5}$ samples collected from two different sites (urban and seaside)
in Xiamen, a coastal city in Southeast China, during spring 2024. Expanding on our
earlier ACSM measurement, OA constituted 30-60% of fine aerosol in Xiamen, with
SOA accounting for over 70% (Chen et al. 2022; Zhang et al. 2020). This work has
three main objectives: (i) to characterize the molecular composition of OA and assess
source impacts; (ii) to compare the physicochemical properties of CHOX compounds
including saturation, oxidation state, and aromaticity, between urban and seaside
environments; and (iii) to elucidate the chemical evolution processes of organic
molecules through case studies. The findings will shed light on the influence of
emission sources and atmospheric chemical processes on OA molecular composition
in different environments.
**2 Experimental Methods**
**2.1 PM$_{2.5}$ Sampling and Offline FIGAERO-I-CIMS Analysis**
The study was conducted at two distinct sampling sites in Xiamen, a city situated
along the southeast coast of China and characterized by a subtropical marine monsoon
climate. The urban site was situated at the Institute of Urban Environment, Chinese
Academy of Sciences (24°26'N, 118°03'E). This site lies in proximity to major roads
(Jimei Avenue and Haixiang Avenue) approximately 100 m away, experiencing high
traffic density. The seaside site was located at the Xiamen Atmospheric Observation
Supersite (24°28'N, 118°10'E), approximately 2.5 km from the coastline and 18.5 km
from the urban site. Potential influence from shipping activities due to the vicinity of
Xiamen Port may affect the seaside site. PM$_{2.5}$ was sampled during the spring season
from 20 March to 30 April, 2024, with a sampling time of 23 hours from 10:00 a.m. to
9:00 a.m. the following day for each sample. A total of 38 and 32 filter samples were
obtained from the urban and seaside sites, respectively, with one procedure blank used
to assess potential contamination during sampling and transportation.
The offline filter sampling steps are similar to previous studies (Hong et al., 2018;



Hong et al., 2022). Briefly, a high-volume aerosol sampler (TH-1000 series, Tianhong
Corp., Wuhan, China) was operated at a flow rate of 1.05 $m^3$ $min^{-1}$ and particulate
matter with a diameter of less than 2.5 μm was collected on pre-baked quartz fiber
filters (18 cm × 23 cm). Before sampling, the quartz filters were wrapped in
aluminum foil and burned in a muffle furnace for 4 h (temperature: 450 ℃) to remove
residual carbon components from the filters. The burned quartz filters were
conditioned in a constant temperature (25 ℃) and humidity (60%) chamber for 24 h,
and then weighed using a balance. After sampling, the filter samples were stored at
-20 ℃ before further chemical analysis. The field blank sample was taken following
the same procedure without drawing air through the sampler.

PM$_{2.5}$ filter samples were analyzed by the FIGAERO-CIMS in offline mode with

negative iodide (I$^-$) ions as the reagent (Aerodyne Research Inc., USA and Tofwerk
AG, Switzerland). Heated and dry ultra-high-purity (UHP) $N_2$ was passed through a
permeation tube containing liquid methyl iodide (CH$_3$I; Alfa Aesar, 99%) to an X-ray
source (Tofwerk AG, P-type), producing I$^-$ to charge the thermally desorbed
compounds. Different from the sandwich method used in other studies (Cai et al.,
2022; Cai et al., 2023; Xin et al., 2024), an area (1.85 cm$^2$) of the sample filters was
punched and placed manually in the dedicated filter holder of FIGAERO directly.
This larger area of filter membrane was used to enhance mass spectrometry signals
due to the relatively low particle concentration in this study. More information was
described in Text S1. A uniform temperature ramping protocol was applied for all
filters, following four steps: (1) stabilization at 25 °C for 1 min; (2) heating from
25 °C to 200 °C in 24 min; (3) soaking at 200 °C for 15 min; and (4) cooling to 25 °C
within 15 min. Two heating cycles were analyzed for each filter sample to assess the
instrument background (Fig. S1). Meanwhile, parallel experiments were conducted to
examine the effects of sampling and operation, with linear regression slopes ranging
from 0.84 to 1.13 and the correlation coefficients upwards to 0.997 (Fig. S2).
**2.2 Coordinated Observations**

Simultaneous observations of atmospheric species were also carried out at the



same stations during the sampling campaign. The method of $N_2O_5$ concentration detected by online I-CIMS was described by Chen et al. (2024). The concentrations of water-soluble ions, including $Na^+$, $NH_4^+$, $K^+$, $Ca^{2+}$, $Mg^{2+}$, $F^-$, $Cl^-$, $NO_3^{2-}$ and $SO_4^{2-}$, were measured by Ion Chromatography (Metrohm 883). OC and EC concentrations were detected by a Model-4 semi-continuous OC/EC aerosol analyzer (Sunset Laboratory Inc., USA). Black carbon (BC) at a wavelength of 880 nm was determined with a model AE-31 Aethalometer (Magee Co., USA). Additional measurements were obtained from instruments deployed at the sampling sites and nearby national air quality monitoring stations, located approximately 1 km and 2 km from the IUE and XS, respectively. Trace gases, i.e., carbon oxide (CO), ozone ($O_3$), sulfur dioxide ($SO_2$), and nitrogen oxides (NO$x$), were simultaneously measured by gas analyzers (Thermo Fisher Scientific, Waltham, MA, USA). Meteorological parameters, including wind speed (WS), wind direction (WD), temperature (T), and relative humidity (RH), were recorded by automatic weather observation station. Ultraviolet radiation (UV) was determined by a UV radiometer (KIPP & ZONEN, SUV5 Smart UV Radiometer). Comprehensive data for all measured parameters at both sites are summarized in Table S1.

**2.3 Data Process and Analysis**

The TofTools package (Junninen et al., 2010; version 6.11) based on MATLAB (MathWorks Inc.) was used to analyze offline FIGAERO-I-CIMS data. The majority of detected compounds fell within the m/z range of 200-500. These ions formed $I^-$ adducts, i.e., $[M]I^-$, where M represents the original molecular formula of the analytes. Besides, a small proportion of compounds existed in other forms, such as losing a hydrogen (H) atom and combining with $NO_3^-$ or $HNO_3I^-$. Mass calibration was performed using five calibrants: $NO_3^-$ (m/z 61.99), $I^-$ (m/z 126.91), $H_2OI^-$ (m/z 144.92), $HNO_3I^-$ (m/z 189.90), and $I_3^-$ (m/z 380.71).

High-resolution peak fitting was performed on 1-minute averaged data, the signal intensity of each compound was subsequently normalized to the signal of reagent $I^-$ and $H_2OI^-$. Signals of the first 1-min of ramping and the last 1-min of



soaking periods were excluded in order to minimize potential interference from
temperature transitions (Cai et al., 2023), the calculation formula of the normalized
signal for each compound i was as follows:
$$Normalized\_Signal_i = \frac{\int_{t=2}^{t=39} signal_i}{\int_{t=2}^{t=39} signal_{I^-} + \int_{t=2}^{t=39} signal_{H_2OI^-}} - \frac{\int_{t=57}^{t=99} signal_i}{\int_{t=57}^{t=94} signal_{I^-} + \int_{t=57}^{t=94} signal_{H_2OI^-}} \quad (1)$$

The detected molecules were classified into four categories based on their

elemental composition: CHO (containing only C, H, and O elements), CHON
(containing 1–2 N atoms in addition to C, H, and O elements), CHOS (containing
only C, H, O, and S elements), and CHONS compounds (containing C, H, O, N, and
S elements). In this study, CHOS and CHONS were collectively categorized as
S-containing compounds. The sum of the four compound classes was denoted as
CHOX, where X represents the possible presence of N, S, or both.

To characterize the properties of the OA compounds, the double bond equivalent

(DBE), oxidation state of carbon atoms ($OS_C$), modified aromaticity index ($AI_{mod}$),
and the effective number of oxygen atoms ($O_{eff}$) were calculated for the obtained
molecular formulas. The DBE, which quantifies the number of rings and double
bonds in a molecule (Koch and Dittmar, 2006), was calculated using Equation (2):
$$DBE = \frac{2 \times c + 2 - h + n}{2} \quad (2)$$

The $OS_C$ was estimated according to Equation (3) described by Kroll et al.

(2011):

$$OSc \approx 2 \times o/c - h/c \quad (3)$$

The $AI_{mod}$ was first proposed by Koch and Dittmar (2006, 2016) to evaluate the

aromaticity of compounds identified through high-resolution mass spectrometry. The
index is calculated based on two parameters. $DBE_{AI}$ is the DBE of molecular core
structure, and $C_{AI}$ is the number of carbon atoms in the core structure. $AI_{mod}$ is defined
as the $DBE_{AI}$ to $C_{AI}$ ratio, with the $AI_{mod} = 0$ when either $DBE_{AI} \leq 0$ or $C_{AI} \leq 0$ (Brege
et al., 2018):



$$\mathrm{AI_{mod}} = \frac{DBE_{AI}}{C_{AI}} = \frac{1+c-0.5\times o - s - 0.5\times(h+n)}{c-0.5\times o - s - n} \quad (4)$$

The $O_{eff}$ was evaluated following Equation (5):
$$\mathrm{O_{eff}} = o - 2\times n - 3\times s \quad (5)$$

In the aforementioned formulas, c, h, o, n, and s represent the number of C, H, O,
N, and S atoms, respectively, in the molecular formulas of the corresponding
compounds. The ratios DBE/C and $O_{eff}$/C indicate the degree of aromaticity and
oxidation of compounds, respectively.
An intensity-weighted parameter ($P_w$) for CHOX compounds was calculated
using Equation (6):
$$\mathrm{P_W} = \sum (P_i \times I_i) / \sum I_i \quad (6)$$

where P represents the following parameters: DBE, DBE/C, $O_{eff}$/C, OSc, or $AI_{mod}$.
$P_i$ corresponds to the specific parameter value for individual compound i and $I_i$
denotes the signal intensity of compound i. The weighted parameters for both
sampling sites are summarized in Table S2.
**3 Results and Discussion**
**3.1 Overview of the sampling period**
During the sampling period, distinct differences in conventional air pollutants
were observed between the urban and seaside sites. As shown in Table S1,
primary-emitted and traffic-related pollutants like CO and NO$x$ exhibited significantly
higher concentrations at the urban site compared to the seaside site, while an opposite
trend was observed for $SO_2$ concentrations. Additionally, the seaside site displayed
enhanced $O_3$ concentrations and UVB intensity, indicating more favorable conditions
for photochemical reactions. The average OC/EC ratio in $PM_{2.5}$ was 6.7 ± 4.4 at the
seaside site, significantly higher than the urban site value of 4.0 ± 0.8 (t-test, *p*<0.001).
Both sites exceeded the SOA formation indicator value (OC/EC = 2) proposed by
Chow et al. (1996), indicating strong SOA production in the study area, particularly at
the seaside site. In this study, a total of 737 and 768 organic molecules (CHOX) were



identified by FIGAERO-I-CIMS for the urban site and the seaside site, respectively.
Significant correlations were observed between CHOX signal intensities and OC
concentrations at both sites (R = 0.70 and 0.80, Fig. S3), demonstrating the reliability
of the OA molecule detection method in this study. The number of identified
molecules is comparable to that at a rural site in Southeast US (769 organic molecules,
Chen et al., 2020) but lower than those observed in highly industrialized urban areas
such as Beijing (939 molecules, Cai et al., 2022) and Guangzhou (815 molecules, Ye
et al., 2021). These spatial differences highlight the significant influence of
environmental conditions on OA molecular composition.

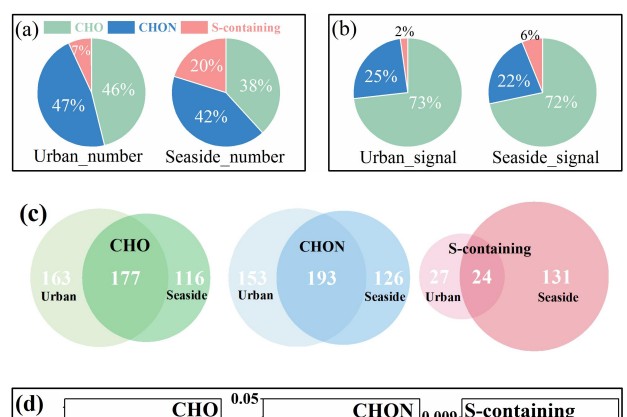

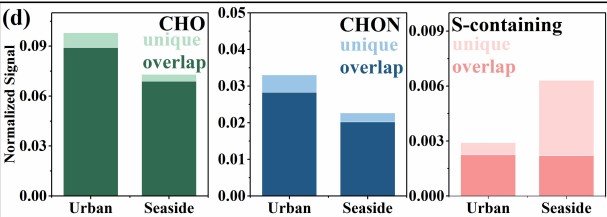

Fig. 1 Numbers (a) and signals intensity (b) proportion of CHO, CHON, and S-containing
compounds in CHOX, Venn diagram of the number distribution of overlapping molecular
formulas between sites (c), and average signals of overlapping and unique CHO, CHON, and
S-containing compounds at both sites (d).
As described in the methodology section, the identified CHOX were classified
into four groups. CHO and CHON compounds were predominant in quantity (Fig. 1a),
accounting for 46% and 47% of the total CHOX at the urban site, and 38% and 42%
at the seaside site, respectively. In terms of signal intensity, CHO and CHON





contributed over 70% and 20% to the total CHOX, respectively, at both sites (Fig. 1b).
This observation is consistent with the well-documented predominance of CHO
compounds across various environments, including urban areas (e.g., 65 ± 5% in
Beijing) (Cai et al., 2022), rural areas (79.9 ± 5.2% on average in the Rhine river
valley, Hyytiälä boreal forest, Finland, and Alabama, US; 87.7 ± 10.8% in Georgia,
US) (Chen et al., 2020; Lopez-Hilfiker et al., 2016) and mountain sites (e.g., 66.2 ±
5.5% in Chacaltaya, Bolivia) (Bianchi et al., 2022). In addition, enhanced
contributions of CHON compounds are consistently found at the urban sites, such as
in Beijing (30 ± 5%) (Cai et al., 2022) and on average across Stuttgart and Karlsruhe,
Germany and Delhi, India (27.1 ± 4.3%) (Haslett et al., 2023; Huang et al., 2019;
Huang et al., 2024), strongly suggesting an association with urban NO$x$ enrichment.
The molecular masses were identified within the range of m/z 200–500 (Fig. S4).
The highest relative abundances, covering m/z 200–320, were found for CHO
compounds, while CHON was mainly concentrated in the range of m/z 320–400.
These characteristics are consistent with the results reported from Beijing, Wuhan,
and Xi'an (Cai et al., 2022; Xin et al., 2024; Shang et al., 2024). As shown in Fig. 1d
and S5, the signal intensities of CHO and CHON were steadily higher at the urban site,
whereas the signals of S-containing compounds were more pronounced at the seaside
site. These differences align with the elevated NO$_2$ at the urban location and the
higher SO$_2$ measured at the seaside location during sampling (Table S1). Comparing
CHOX chemical compounds at the two sites (Fig. 1c and S6), we found that over 50%
of CHO and CHON molecules at both sites shared the same molecular formula,
accounting for 86%–94% of total signals. Notably, the overlapping S-containing
compounds constituted only a minor proportion of total S-containing species at the
seaside site (35%). These results reveal distinct variations in OA composition between
the urban and seaside sites, attributable to differences in anthropogenic emission
influences. To further explore the discrepancies in CHOX compounds between the
two sites, the characteristics and properties of CHOX were analyzed in detail.
**3.2 Characteristics and Properties of CHOX Compounds**



### 3.2.1 The Characteristics of CHOX Distribution


The bulk molecular formulas of CHOX compounds were determined as
$C_{10.8}H_{13.7}N_{0.5}O_{5.4}S_{0.1}$ at the urban site and $C_{10.7}H_{14.4}N_{0.4}O_{5.4}S_{0.2}$ at the seaside site, with mean
weighted effective oxygen numbers ($O_{eff}/C$) of 0.82 and 0.85, respectively, indicating
the highly oxidized nature of these compounds in the coastal region. In comparison,
the bulk CHOX composition at the seaside site exhibited a higher O/C ratio, fewer
CHON compounds, and more S-containing compounds. The elevated CHON signal
intensity at the urban site could be attributed to the high NO$x$ concentrations from
vehicle emissions, facilitating nitrogen-containing compound formation. In contrast,
the enhanced S-containing compound signals at the seaside site likely resulted from
marine-driven sources.
Figure 2 and S7 show the signal intensity and quantity distribution of CHO and
CHON compounds as a function of carbon number. The distribution characteristics of
CHO and CHON compounds were broadly consistent between the two sites. As
shown in Fig. S7, the quantity of CHO compounds exhibited a normal distribution
with carbon number, mainly concentrated in the ranges of 8–14 carbon and 4–6
oxygens, while the distribution of CHON compounds was relatively uniform, with a
minor abundance peak at $C_{6-10}$ and $O_{5-6}$. In contrast to the quantity distribution, the
signal intensity of CHO compounds decreased with increasing carbon number (Fig.
2a–b). Both sites exhibited significant signal contributions from $O_4$-CHO compounds,
which likely correspond to dicarboxylic acid species. Oxalic acid ($C_2$) and succinic
acid ($C_4$) were the most abundant species, followed by malonic acid ($C_3$). These
low-molecular-weight dicarboxylic acids are typically associated with aqueous
reactions (Lim et al., 2010). With increasing carbon number, the dominant oxygen
content in CHO compounds shifted from $O_4$ to $O_{5-6}$. The signals of CHON
compounds were predominantly concentrated in the $C_{3-10}$ range, with their oxygen
content shifting from $O_3$ to $O_{5-7}$ as carbon number increased (Fig. 2c–d). Site-specific
differences in compound distribution were observed. CHO and CHON compounds
with high carbon content ($>C_{10}$) exhibited stronger signal intensities at the urban site



compared to the seaside site, with $O_6$-CHO and $O_8$-CHON species showing particular
enhancement.

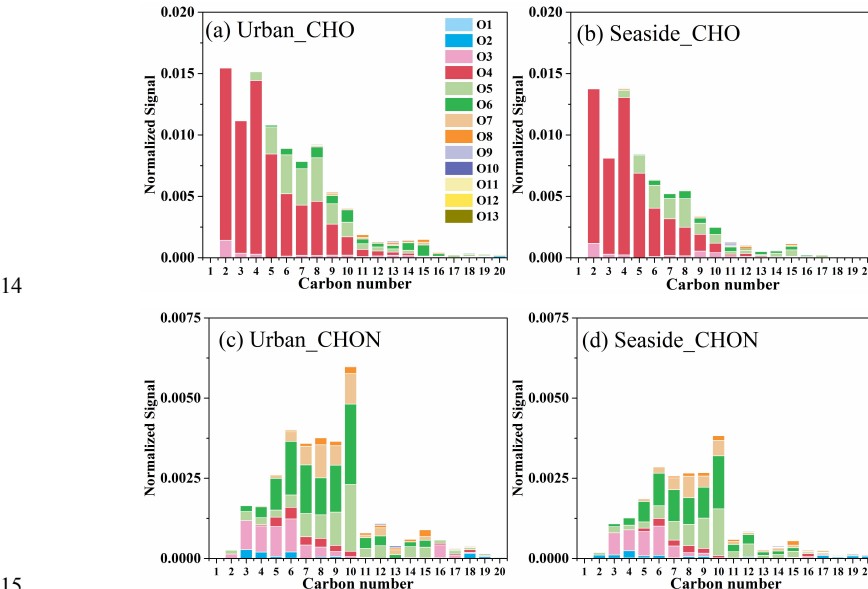



Fig. 2 Signal intensity of CHO and CHON categorized by the number of carbon atoms at the
urban (a, c) and seaside (b, d) sites.
**3.2.2 The Unsaturation, Oxidation State, and Aromaticity of CHOX Compounds**
The double bond equivalent (DBE) characterizes the potential number of rings
and double bonds in organic compound molecules. Quantitatively, CHOX compounds
were predominantly distributed within the DBE range of 2–8 (Fig. S8). In terms of
signal intensity (Fig. 3), CHO compounds exhibited significant contributions in the
DBE = 2–4 range, while CHON compounds were concentrated in the DBE = 2–5
range. Compounds with DBE ≥ 6 showed higher signal contributions at the urban site
(9%) than at the seaside site (7%). These highly unsaturated compounds may undergo
oxidative transformation into higher molecular weight products, being particularly
prone to photooxidation with oxidants like $O_3$-containing species to form of C=O
bonds in carbonyls and carboxylic acids (Zhao et al., 2014). As shown in Table S2, the
weighted-DBE values of CHOX compounds ranged from 2.32 to 3.68, which is close
to the previously reported values (2.64–3.82) for urban and marine samples (Xin et al.,



2024). The urban site exhibited higher weighted-DBE values for CHOX compounds
(3.25) compared to the seaside site (2.99) (Table S2), indicating distinct formation
pathways. The highly unsaturated CHOX compounds likely derived from
anthropogenic precursors such as aromatic VOCs and PAHs (Chen et al., 2014),
whereas the more saturated components primarily originated from biogenic terpene
compounds (Du et al., 2024; Chan et al., 2011; Nguyen et al., 2012; Noziere et al.,
2010).

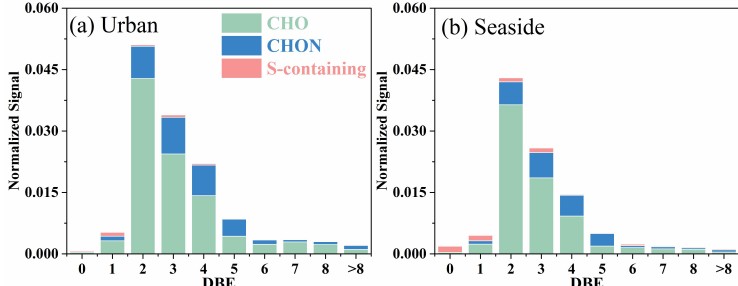

Fig. 3 Signal intensity of CHOX categorized by double bonds equivalent (DBE) at the urban site

(a) and the seaside site (b).

DBE/C ratios greater than 0.7 generally indicate the presence of soot materials or
oxidized polycyclic aromatic hydrocarbons (PAHs) (Cui et al., 2019). During the
sampling period, while the signal proportions of compounds with DBE/C > 0.7 were
comparable between the two sites (CHO: ~25%; CHON: ~16%), their number
proportions were significantly higher at the urban site (CHO: 17%; CHON: 22%) than
at the seaside site (CHO: 6%; CHON: 19%). Consistent with the above discussion,
S-containing compounds with DBE/C > 0.7 exhibited higher proportions in both
signal intensity and quantity at the urban site (24% and 26%, respectively) compared
to the seaside site (14% and 10%, respectively). The systematically higher DBE/C
ratios for OA compounds at the urban site suggest more contribution from soot
materials and/or oxidized PAHs, which aligns with the observed EC concentrations
(urban: $1.57 \pm 0.80$ µg m$^{-3}$; seaside: $0.91 \pm 0.42$ µg m$^{-3}$). The AI$_{mod}$, which reflects the
minimum number of carbon-carbon double bonds and aromatic rings (Koch and
Dittmar, 2006, 2016), was correspondingly higher at the urban site





(weighted-$AI_{mod}$=0.17) than at the seaside site (0.15), further supporting the greater
contribution of aromatic species to urban OA.

The carbon oxidation state (OSc), a parameter introduced by Kroll et al. (2011),

serves to quantify the oxidation degree of organic mixtures undergoing dynamic
atmospheric processes. The weighted-OSc values of CHOX compounds were higher
at the seaside site (0.55) than at the urban site (0.49) (Table S2), consistent with the
aforementioned enhanced SOA formation at the seaside site. The van Krevelen (VK)
diagrams (Fig. S9) revealed that homologue series such as $C_nH_{2n-x}O_4$ (where x=2,4,6)
made significant contributions to CHOX compounds in this study. The elevated
weighted-OSc values of seaside CHO compounds could be attributed to the strong
signal intensities of low-carbon-number homologues, such as $C_nH_{2n-2}O_4$ and $C_nH_{2n-4}O_4$.
In contrast to CHO compounds, both CHON and S-containing compounds exhibited
higher weighted-OSc values at the urban site, indicating a more chemically aged state
of N- and S-containing species. The increased weighted-OSc of urban CHON
compounds mainly resulted from enhanced signals of highly oxidized species, such as
$C_3H_5NO_6$ and $C_5H_7NO_6$. The relatively low weighted-OSc of seaside S-containing
compounds was most likely associated with fresh sulfur emissions from local sources,
including oceanic discharge and ship exhaust. This observation aligns with previous
findings that S-containing compounds in primary organic sea spray aerosols are
predominantly composed of fatty acids and other lipid molecules with lower oxidation
degrees (Siegel et al., 2021). Additionally, the limited number of identified
S-containing compounds may have artificially inflated the weighted-OSc values for
urban S-containing compounds.
**3.2.3 The Classification of CHOX Compounds**

As shown in Fig. 4, CHOX compounds were categorized based on $AI_{mod}$, DBE,

H/C, and O/C ratios. Low H/C ratios, combined with high DBE and $AI_{mod}$ values
indicate a high degree of unsaturation and the presence of aromatic structures. CHOX
were classified into Aromatic-like and Aliphatic-like compounds based on $AI_{mod}$ and
H/C, following the methodology of Xin et al. (2024). Compounds with $AI_{mod}$ > 0.5 or



AI$_{mod}$ ≤ 0.5 and H/C < 1.5 were defined as Aromatic-like compounds. Conversely,
compounds with AI$_{mod}$ ≤ 0.5 and H/C ≥ 1.5 were defined as Aliphatic-like compounds
(Coward et al., 2019). The categorical distribution of CHOX was generally consistent
between the two sites, with Aromatic-like CHOX dominating over Aliphatic-like
CHOX in signal intensity, indicating prevalent aromatic precursor contributions to OA
formation across the study area.

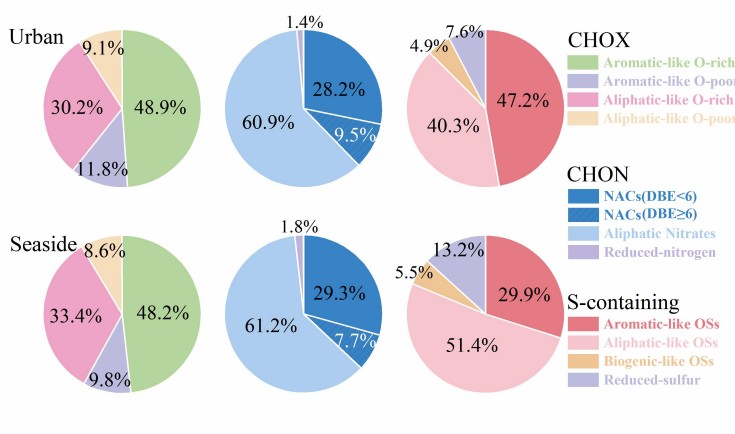


Fig. 4 Fraction of CHOX signal intensity categorized by different parameter.
Using an O/C threshold of 0.5, both Aromatic-like and Aliphatic-like CHOX
were further subdivided into two subcategories, with O-rich (O/C > 0.5) compounds
being more abundant than O-poor (O/C ≤ 0.5) compounds. The signal ratios of O-rich
to O-poor compounds were higher at the seaside site for both categories. Specifically,
the O-rich/O-poor ratio for Aromatic-like CHOX was 4.9 at the seaside site versus 4.1
at the urban site. Similarly, the ratio for Aliphatic-like CHOX was 3.9 at the seaside
site compared to 3.3 at the urban site. These results are consistent with the findings of
OC/EC ratios and OSc discussed earlier, suggesting that the seaside atmosphere is
more conducive to the formation of highly oxidized organic compounds. Notably, the
proportion of Aliphatic-like O-rich CHOX was significantly higher at the seaside site
than at the urban site, aligning with a recent study that reported elevated levels of
Aliphatic-like O-rich CHOX in marine-remote PM$_{2.5}$ (Xin et al., 2024). These
findings demonstrate a consistent spatial pattern that urban OA is more influenced by



anthropogenic emissions, dominated by aromatic species, whereas marine-influenced
OA exhibits higher proportions of aged, aliphatic compounds.
For CHON compounds, Aromatic-like compounds with O/N ≥ 2 and
Aliphatic-like compounds with O/N ≥ 3 were categorized as Nitro-aromatic
compounds (NACs) and Aliphatic Nitrates, respectively, while the remaining
compounds were classified as Reduced-nitrogen compounds. The distribution patterns
of CHON classifications were highly consistent between the two sites. Aliphatic-like
CHON exhibited the highest signal proportion, followed by NACs, with
Reduced-nitrogen compounds showing the lowest signal contribution. The most
abundant compound in Aliphatic-like CHON was $C_{10}H_{15}NO_6$, accounting for 7.5% of
total signals at both sites. Previous studies have demonstrated that $C_{10}H_{15}NO_6$ is
produced through nitrate radical ($NO_3\cdot$)-initiated oxidation of monoterpenes (e.g.,
limonene or β-pinene, $C_{10}H_{16}$) (Boyd et al., 2015; Faxon et al., 2018). Regarding
NACs, $C_nH_{2n-7}O_xN$ homologues (e.g., $C_6H_5NO_3$, $C_8H_{11}NO_7$, and $C_7H_7NO_3$), likely
originating from vehicle or combustion emissions, dominated the signal profiles at
both sites. Notably, NACs with DBE ≥ 6 exhibited higher signal intensities at the
urban site, primarily attributable to stronger influences from traffic- or
combustion-related sources.
Following Lin et al. (2012) and Xie et al. (2020), S-containing compounds with
O/S ≥ 4 were defined as organosulfates (OSs) and further divided into three
subcategories: (1) Aromatic-like OSs ($AI_{mod}$ > 0.5, or $AI_{mod}$ ≤ 0.5 with H/C < 1.5); (2)
Aliphatic-like OSs ($AI_{mod}$ ≤ 0.5, H/C ≥ 1.5, and DBE ≤ 2); (3) Biogenic-like OSs
($AI_{mod}$ ≤ 0.5, H/C ≥ 1.5, and DBE > 2). Compounds with O/S < 4 were classified as
Reduced-sulfur compounds. Unlike CHO and CHON, the classification distribution of
S-containing compounds showed marked differences between the two sites. At the
urban site, Aromatic-like OSs dominated the signal profiles, followed by
Aliphatic-like OSs. In contrast, the seaside site exhibited the highest signal proportion
of Aliphatic-like OSs, primarily contributed by compounds such as $C_3H_6O_5S$,
$C_3H_8O_5S$, and $C_4H_{10}O_5S$. Additionally, the seaside site showed higher signal



intensities of Reduced-sulfur compounds, primarily owing to the identification of
abundant $C_nH_{2n-4}O_xS$ and $C_nH_{2n+2}O_xS$ homologues.

**3.3 Case study: Evolution of molecular compositions**

To investigate the impact of atmospheric processes on organic molecular
composition, we selected two distinct episodes with significant increases in $PM_{2.5}$
concentrations (Case 1: March 26–29; Case 2: April 10–13) for further analysis (Fig.
S10). In Case 1, the daily average offline $PM_{2.5}$ concentration increased from 27.35 to
38.80 µg m$^{-3}$ at the urban site and from 27.37 to 45.89 µg m$^{-3}$ at the seaside site. While
in Case 2, it increased from 17.13 to 59.78 µg m$^{-3}$ at the urban site and from 17.11 to
40.63 µg m$^{-3}$ at the seaside site. Backward trajectory analysis (Fig. S11) revealed that
the air masses in Case 1 were transported from North China along the coastline to
study area over a long distance and then shifted to local air masses. This transport
pattern was accompanied by initially elevated nighttime $O_3$ concentrations under
regional influence, followed by a significant rise in NO$x$ levels due to local
accumulation. RH in Case 1 remained relatively stable at both sites, averaging
approximately 72% (urban) and 86% (seaside). In contrast, the air masses in Case 2
were primarily influenced by long-range marine transport. The O$x$ levels displayed
repetitive and stable diurnal variations. Notably, RH progressively increased from
66% to 91% at the urban site and from 76% to 99% at the seaside site, while UVB
intensity declined gradually from 12.6 to 5.4 w m$^{-2}$ at the urban site and from 12.4 to
8.7 w m$^{-2}$ at the seaside site. We also observed that as Case 2 progressed, the $NO_3^-$ and
$SO_4^{2-}$ concentrations increased markedly with rising RH. Overall, the two episodes
exhibited distinct source origins and environmental conditions.
The evolution of daily CHOX signal intensity during pollution episodes is
presented in Fig.5. Case 1 exhibited higher CHOX signal intensities than Case 2.
Spatially, the urban site showed higher levels of CHO and CHON compounds, while
the seaside site had relatively higher concentrations of S-containing compounds,
consistent with the general characteristics of the two sites. During Case 1, all CHOX
compounds showed increased signal intensities, particularly urban CHON and seaside



S-containing compounds (Fig. 5b–d). This feature aligns with the earlier hypothesis
that local emissions dominantly contributed to CHOX in this episode. As expected,
NO$x$ concentrations were significantly higher at the urban site (38.21 μg m$^{-3}$) than at
the seaside site (24.91 μg m$^{-3}$), while SO$_2$ levels exhibited an inverse trend (urban:
3.29 μg m$^{-3}$; seaside: 4.54 μg m$^{-3}$). In contrast, Case 2 (Fig. 5b–d) was characterized
by a significant temporal increase in CHO compounds, while CHON levels remained
stable. Additionally, S-containing compounds showed a slight but synchronized
enhancement at both sites, likely due to the influence of a common marine air mass
source. Given the more pronounced variations in CHOX composition at the urban site,
we conducted further analysis on CHOX at this location.

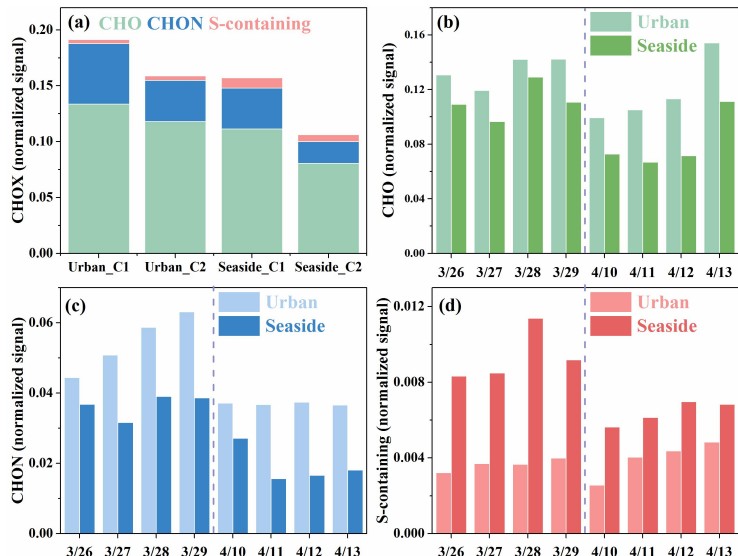


Fig. 5 Signal intensity of CHOX during the different periods at the two sites.

Figure 6 illustrates the compositional changes of CHOX compounds during the

two pollution episodes. The OSc (0.37) of CHOX in Case 1 was significantly lower
than the campaign average (0.49). Regarding CHOX composition, as the dominant
influence shifted from long-distance transport to local accumulation, the proportion of
Aliphatic-like O-rich components in CHOX decreased, while Aliphatic-like O-poor
components contributed increasingly. These observations reflect the enhanced
influence from local anthropogenic emissions. Our aforementioned results have



shown a significant enhancement of CHON signals in Case 1. Further compositional

analysis revealed that as the episode progressed, the proportion of NACs (DBE<6) in

CHON compounds decreased, accompanied by a modest increase in Aliphatic

Nitrates. Figures 7a-d (marked by blue dashed boxes) specifically show that the urban

CHON signal enhancement was primarily driven by low DBE/C compounds,

including $C_{10}H_{19}NO_5$, $C_{10}H_{15}NO_6$, $C_4H_7NO_3$, $C_6H_{11}NO_6$, and $C_3H_7NO_3$. These

organonitrates can be formed through daytime reactions between NO and organic

peroxy radicals (generated from monoterpene oxidation by OH radicals or $O_3$), or via

nighttime $NO_3$ radical-initiated reactions with monoterpenes (Atkinson and Arey,

2003; Kurten et al., 2017; Yan et al., 2016). During Case 1, elevated $O_3$ concentrations

resulting from regional transport (Fig. S10a) likely facilitated nocturnal $NO_3$ radical

formation via reactions with locally emitted $NO_2$. Online CIMS measurements also

revealed that $N_2O_5$ signals in Case 1 were one order of magnitude higher than in Case

2 (data not shown). Together, these findings suggest that $NO_3$ radical-initiated

nighttime reactions promoted the formation of Aliphatic Nitrates (e.g., $C_{10}H_{19}NO_5$ and

$C_{10}H_{15}NO_6$) in CHON compounds during this stagnation events.

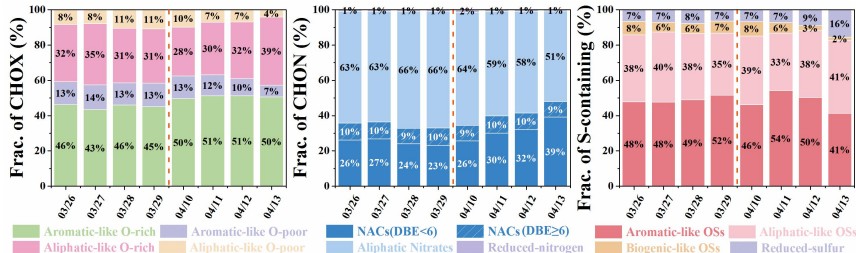

Fig. 6 Fraction distribution of urban CHOX signal intensity categorized by different parameter

($AI_{mod}$, DBE,H/C, and O/C) during Case 1 and Case 2.

Case 2 exhibited distinct characteristics compared to Case1 (Fig. 6). The OSc

(0.51) of CHOX compounds in Case 2 was increased in contrast to the campaign

average (0.49). The proportion of Aliphatic-like O-rich compounds demonstrated a

continuous upward trend, rising from 28% to 39% at the urban site, while

Aromatic-like O-rich compounds remained stable. Conversely, the proportion of both

Aromatic-like O-poor and Aliphatic-like O-poor compounds decreased. Unlike CHO



compounds, the total signal intensities of CHON compounds in Case 2 remained relatively constant (Fig. 5c), but their molecular composition underwent significant changes. Specifically, the proportion of NACs (DBE<6) in CHON compounds increased from 26% to 39% at the urban site, whereas Aliphatic Nitrates decreased correspondingly from 64% to 51%. Additionally, variations in S-containing compounds were generally consistent across both sites (Fig. S12), with the most notable increases observed in Aliphatic-like OSs and Reduced sulfur compounds.

A distinct characteristic of Case 2 was the increased signal intensity of CHO compounds (Fig. 5b), particularly on April 13. The VK diagrams (Fig. 7i–l) showed that CHO compounds evolved into highly oxidized species (e.g., $C_2H_2O_4$, $C_4H_6O_4$, $C_5H_8O_4$) as aerosol concentrations increased. Previous studies have demonstrated that organic acids are continuously generated during aqueous photochemical aging processes (Ervens et al., 2011; Ye et al., 2025). The $C_nH_{2n-2}O_4$ homologues, such as $C_4H_6O_4$ (succinic acid) and $C_3H_4O_4$ (malonic acid), were predominantly formed via aqueous-phase reactions and/or photochemical oxidation of VOCs (Kawamura et al., 2016). The $C_5H_8O_4$ species likely represents glutaric acid, a known oxidation product of isoprene (Berndt et al., 2019). Research indicates that aqueous-phase reactions play a crucial role in generating highly oxidized OA (Xu et al., 2017b). We therefore infer that the increased RH under marine air mass influence during Case 2 promoted aqueous reaction-driven organic aerosol formation, thereby enhancing OA oxidation levels. This interpretation is supported by the observation that CHO compounds at the seaside site exhibited higher oxidation degrees than those at the urban site (Table S2). Regarding CHON compositional changes, the VK diagrams clearly showed a decline in signals from low DBE/C species (e.g., $C_{10}H_{19}NO_5$, $C_{10}H_{15}NO_6$, and $C_{10}H_{17}NO_6$, blue dashed box in Fig. 7e-h) alongside an increase in oxygen-rich and lower-carbon species (e.g., $C_6H_5NO_3$, $C_5H_3NO_3$, $C_3H_5NO_6$, $C_5H_5NO_6$, and $C_5H_9NO_6$, orange dashed box in Fig. 7e-h). These CHON compounds most likely originated from aromatic hydrocarbon oxidation followed by atmospheric aging processes, leading to concurrent increases in both weighted-$AI_{mod}$ (0.16 vs. 0.14) and weighted-$OSc$ (0.02




vs. -0.08). Overall, the compositional evolution in Case 2 demonstrates that
marine-derived humid air masses enhanced aqueous-phase reactions, thereby
promoting organic aerosol formation and intensifying oxidative states.

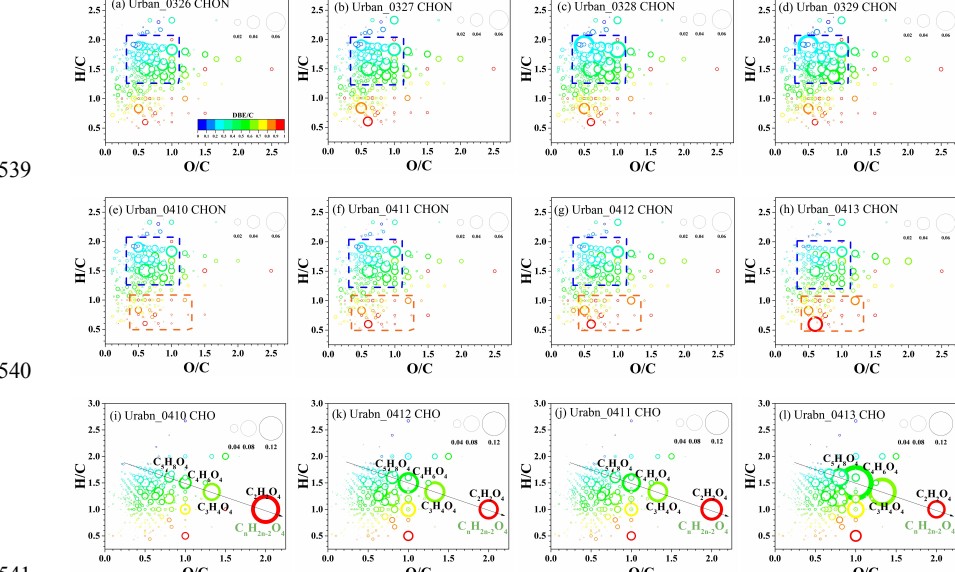




Fig. 7 The van Krevelen (VK) diagram of urban CHON (a–h) and CHO (i–l) day by day during
different periods. The circle size corresponds to signal intensity and the color scale represents the
DBE/C ratio. The low DBE/C ratio species are marked in the blue dashed box. The oxygen-rich

and lower-carbon species are marked in the orange dashed box.

**4 Conclusions**
This study investigates the molecular characteristics and chemical evolution of
OA in coastal environments (urban and seaside sites) through FIGAERO-I-CIMS
analysis of $PM_{2.5}$ samples collected during spring 2024. CHO and CHON compounds
dominated the OA composition at both sites, sharing over 50% of molecular formulas
and accounting for 86%–94% of total signal intensities. The urban site exhibited
higher signal intensities of CHON compounds, while the seaside site showed elevated
S-containing compounds. These results clearly reflect distinct source-specific
molecular fingerprints. The $O_{eff}/C$ values (urban 0.82; seaside 0.85) indicated higher
oxidation levels in coastal CHOX compounds. Compared to urban OA, Seaside OA



exhibited lower unsaturation, reduced aromaticity, and higher oxidation states.
Categorization showed that Aromatic-like CHOX exhibited higher signals than
Aliphatic-like compounds at both sites, while urban OA was enriched in aromatic
species (e.g., NACs from vehicle- and combustion-related emissions) and seaside OA
featured aliphatic and highly oxidized compounds. Two pollution episodes were
selected to investigate CHOX evolution mechanisms. Case 1 (local accumulation)
exhibited a significant increase in urban CHON compounds, likely resulting from
$NO_3\cdot$-initiated nighttime oxidation that promoted the formation of Aliphatic Nitrates
under local NO$x$ accumulation. Case 2 (marine air masses) showed increased
proportions of Aliphatic-like O-rich CHOX compounds (28% to 39%) via
aqueous-phase processing under high humidity. The offline FIGAERO-I-CIMS
proved robust for molecular-level characterization of OA across diverse environments.
These findings not only advance our understanding of OA molecular characteristics
and chemical evolution processes, but also provide insights for region-specific control
strategies.

**CRediT authorship contribution statement**
YC conducted the laboratory experiments. YC and LX analyzed the data and
wrote the paper. ZL, CY, GC, RZ, and YZ coordinated the measurements and
maintained data. LX, XF, and JC designed the project. JC, YH, and ML supervised the
study. All the co-authors contributed to the discussion and commented on the
manuscript.
**Declaration of Competing Interest**
The authors declare that they have no known competing financial interests or
personal relationships that could have appeared to influence the work reported in this
paper.
**Data availability**
The data related to this article are accessible at figshare
(https://doi.org/10.6084/m9.figshare.28956629, Chen et al., 2025).



**Acknowledgements**
This study was funded by the National Natural Science Foundation of China
(U22A20578 and 42277091), the National Key Research and Development Program
(2022YFC3700304), the Science and Technology Department of Fujian Province
(2022L3025), the guiding project of seizing the commanding heights of
"self-purifying city" (no. IUE-CERAE-202402), STS Plan Supporting Project of the
Chinese Academy of Sciences in Fujian Province (2023T3013), and Xiamen
Atmospheric Environment Observation and Research Station of Fujian Province.

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
