# Peer review of "Measurement report: Molecular characterization of organic aerosol"

_EGUsphere, 2025_

## Author Comment (AC1)

**Responses to comments from the reviewer 1**

Revised Review Comments:

The paper by Chen et al. provides a clear analysis of the molecular characterization of organic aerosols in urban and seaside environments within a coastal city in southeast China. The dataset is credible and thoroughly analyzed, enhancing our understanding of molecular-level organic aerosol characteristics in coastal areas. The manuscript is generally well written, though it contains several English errors that require attention. I recommend acceptance after minor revisions, as detailed below:

**Response:** We appreciate your overall positive assessment of the manuscript. We have thought about the issues you raised and carefully revised the entire manuscript accordingly. Detailed responses are listed below, and the main correlations are marked in blue in the "Revised manuscript with changes marked" file.

Lines 116-117: Rewrite the sentence to:

"Potential influence from shipping activities, due to the proximity of Xiamen Port, may impact the seaside site."

**Response:** We rewrote the site introduction, and this sentence has been modified. (in lines 117–122)

Line 149: Clarify what the correlation coefficients represent.

**Response:** We rewrote the sentence in lines 152–156. "Meanwhile, parallel experiments were conducted to evaluate the reproducibility of sampling and analytical procedures. These tests showed excellent agreement between the OA signal intensities of duplicate samples, with linear regression slopes of 0.84–1.13 and correlation coefficients ($r^2$) upwards to 0.997 (Fig. S2)."

Lines 157-158: The BC-related data is not used in the article and could be deleted.

**Response:** We have corrected it.

Lines 190-191: Rewrite the sentence as:

"The sum of the four compound classes was denoted as CHOX, where X indicates the potential presence of N, S, or both."

**Response:** We rewrote the sentence in lines 195–196.

Line 228: Ensure consistent terminology for "$O_3$ concentrations and UVB intensity."

**Response:** We have corrected it. And we ensured the consistency of terminology throughout the full text.

Lines 343-345: "The signal proportion of compounds with DBE/C > 0.7 was comparable between the two sites (CHO: ~25%; CHON: ~16%)…"

Question: Were low-carbon-number compounds (e.g., $C_2H_2O_4$) excluded from this analysis?

**Response:** The low-carbon-number compounds were not excluded from the dataset. We calculated the signal proportion of compounds with DBE/C > 0.7 after excluding $C_2H_2O_4$, which was 10% for the urban site and 8% for the seaside site. This indicates that regardless of whether small-molecule compounds are excluded, the proportion of compounds with DBE/C > 0.7 remains higher at the urban site than at the seaside site. The results demonstrate that urban OA compounds systematically exhibit higher DBE/C ratios compared to those at the seaside site.

Lines 417-420: "…urban OA is more influenced by anthropogenic emissions, dominated by aromatic species, whereas marine-influenced OA exhibits higher proportions…"

Note: Verbs should be in singular form for consistency.

**Response:** We have corrected it. We rewrote the sentence combined with other comments. "This finding demonstrates a consistent spatial pattern that urban OA is more strongly influenced by anthropogenic emissions and is dominated by aromatic species, whereas marine-influenced OA exhibits relatively higher proportions of aged,

aliphatic compounds." (in lines 423–427)

Lines 434-435: "…owing to the identification of abundant $C_nH_{2n-4}O_xS$ and $C_nH_{2n+2}O_xS$ homologues…"

Suggestion: Provide additional context on gas-phase contributions.

**Response:** Thanks for your suggestion. The abundant reduced sulfur ($C_nH_{2n-4}O_xS$ and $C_nH_{2n+2}O_xS$, x<4) might be generated from the oxidation of dimethyl sulfide (DMS) emitted by marine phytoplankton. In addition, fatty acids and other lipid molecules carried in primary organic sea spray aerosols can react with other organic compounds to form the above-mentioned Aliphatic-like OSs. We have added explanations regarding the sources and formation pathways of these S-containing compounds in the corresponding text.

"The elevated abundances of Aliphatic-like OSs and Reduced-sulfur compounds in seaside OA were likely attributed to the oxidation of biogenic reduced sulfur gases, particularly dimethyl sulfide (DMS) emitted from the ocean (Shen et al., 2022; Siegel et al., 2021)." (in lines 463–466)

Additional references:

Shen, J., Scholz, W., He, X.-C., Zhou, P., Marie, G., Wang, M., Marten, R., Surdu, M., Rörup, B., Baalbaki, R., Amorim, A., Ataei, F., Bell, D. M., Bertozzi, B., Brasseur, Z., Caudillo, L., Chen, D., Chu, B., Dada, L., Duplissy, J., Finkenzeller, H., Granzin, M., Guida, R., Heinritzi, M., Hofbauer, V., Iyer, S., Kemppainen, D., Kong, W., Krechmer, J. E., Kürten, A., Lamkaddam, H., Lee, C. P., Lopez, B., Mahfouz, N. G. A., Manninen, H. E., Massabò, D., Mauldin, R. L., Mentler, B., Müller, T., Pfeifer, J., Philippov, M., Piedehierro, A. A., Roldin, P., Schobesberger, S., Simon, M., Stolzenburg, D., Tham, Y. J., Tomé, A., Umo, N. S., Wang, D., Wang, Y., Weber, S. K., Welti, A., Wollesen de Jonge, R., Wu, Y., Zauner-Wieczorek, M., Zust, F., Baltensperger, U., Curtius, J., Flagan, R. C., Hansel, A., Möhler, O., Petäjä, T., Volkamer, R., Kulmala, M., Lehtipalo, K., Rissanen, M., Kirkby, J., El-Haddad, I., Bianchi, F., Sipilä, M., Donahue, N. M., and Worsnop, D. R.: High Gas-Phase Methanesulfonic Acid Production in the OH-Initiated Oxidation of Dimethyl Sulfide at Low Temperatures, Environ. Sci. Technol., 56, 13931-13944. doi:10.1021/acs.est.2c05154, 2022.

Lines 488-490: "…reactions of NO with organic peroxy radicals (generated from monoterpenes oxidation by OH radicals or $O_3$) during the daytime, or via nighttime

NO$_3$ radical-initiated reactions…”

Question: Were aqueous/heterogeneous phase reactions considered?

**Response:** Indeed, aqueous and heterogeneous phase reactions are critical for the formation of organic nitrates in the aerosol phase. Currently proposed major formation pathways for organic nitrates include the addition of nitrate radicals (NO$_3$·) to C=C bonds and the reactions of organic peroxyl radicals with nitric oxide (NO). Particulate organic nitrates are primarily generated by gas-particle partitioning of gaseous organic nitrates and the heterogeneous reaction of NO$_3$· with organic aerosols (Yang et al., 2025). Our initial analysis focused predominantly on gas-phase oxidation. We have revised the manuscript to explicitly address these multiphase formation processes.

“These organonitrates are mainly formed through the oxidation of VOCs such as alkanes, alkenes and monoterpenes by OH, O$_3$, and NO$_3$ in the presence of NO$x$ (Lee et al., 2016; Ng et al., 2017; Yan et al., 2019; Yang et al., 2025), and subsequently partition into the particle phase. Additionally, particulate organonitrates can also be generated via heterogeneous reactions of organic compounds with NO$_3$ (Nah et al., 2016).” (in lines 518–523)

Additional references:

Lee, B. H., Mohr, C., Lopez-Hilfiker, F. D., Lutz, A., Hallquist, M., Lee, L., Romer, P., Cohen, R. C., Iyer, S., Kurtén, T., Hu, W., Day, D. A., Campuzano-Jost, P., Jimenez, J. L., Xu, L., Ng, N. L., Guo, H., Weber, R. J., Wild, R. J., Brown, S. S., Koss, A., Gouw, J. de, Olson, K., Goldstein, A. H., Seco, R., Kim, S., McAvey, K., Shepson, P. B., Starn, T., Baumann, K., Edgerton, E. S., Liu, J., Shilling, J. E., Miller, D. O., Brune, W., Schobesberger, S., D'Ambro, E. L., and Thornton, J. A.: Highly functionalized organic nitrates in the southeast United States: Contribution to secondary organic aerosol and reactive nitrogen budgets. P. Natl. Acad. Sci. USA, 113, 1516-1521. doi:10.1073/PNAS.1508108113, 2016.

Nah, T., Sanchez, J., Boyd, C. M., and Ng, N. L.: Photochemical Aging of α-pinene and β-pinene Secondary Organic Aerosol formed from Nitrate Radical Oxidation. Environ. Sci. Technol., 50, 222-231. doi:10.1021/acs.est.5b04594, 2016.

Ng, N. L., Brown, S. S., Archibald, A. T., Atlas, E., Cohen, R. C., Crowley, J. N., Day, D. A., Donahue, N. M., Fry, J. L., Fuchs, H., Griffin, R. J., Guzman, M. I., Herrmann, H., Hodzic, A., Iinuma, Y., Jimenez, J. L., Kiendler-Scharr, A., Lee, B. H., Luecken, D. J., Mao, J., McLaren, R., Mutzel, A., Osthoff, H. D., Ouyang, B., Picquet-Varrault, B., Platt, U., Pye, H.

O. T., Rudich, Y., Schwantes, R. H., Shiraiwa, M., Stutz, J., Thornton, J. A., Tilgner, A., Williams, B. J., and Zaveri, R. A.: Nitrate radicals and biogenic volatile organic compounds: oxidation, mechanisms, and organic aerosol. Atmos. Chem. Phys., 17, 2103-2162. doi:10.5194/acp-17-2103-2017, 2017.

Yang, Y., Huang, L., Zhao, M., Wu ,Y., Xu, Y., Li, Q., Wang, W., and Xue, L.: Multiphase reactions of organic peroxides and nitrite as a source of atmospheric organic nitrates. Nat. Commun.. 16, 5437. doi:10.1038/s41467-025-60696-3, 2025.

Lines 493-495: "…$N_2O_5$ signals in Case 1…"

Question: How does $O_3$ concentration compare in Case 1? Is it lower?

**Response:** The $O_3$ concentration in case 1 showed a higher value under the influence of regional transport, which led to the formation of more $NO_3$ radicals by $NO_2+O_3$. As we mentioned in the text, "During Case 1, elevated $O_3$ concentrations resulting from regional transport (Fig. S10a) likely facilitated nocturnal $NO_3$ radical formation via reactions with locally emitted $NO_2$."

---

## Author Comment (AC2)

**Responses to comments from the reviewer 2**

The manuscript by Chen et al. investigates the molecular composition and potential chemistry processes of organic aerosol samples collected from urban and seaside sites of a coastal environment with the deployment of a FIGAERO-CIMS mass spectrometer. The manuscript is well written and the topic of this manuscript is interesting. However, some revisions with more discussions are needed before its possible publication on ACP. Please see my comments and questions below.

**Response:** We appreciate your overall positive assessment of the manuscript. We have thought about the issues you raised and carefully revised the entire manuscript accordingly. First, we would like to provide further clarification regarding the sampling sites, which has also been elaborated in the site description section in the revised manuscript (in lines 117–122). This study was conducted at two sampling sites (an urban site and a seaside site) in a coastal city. Both sites are affected by anthropogenic and oceanic sources, but to varying degrees. Specifically, in terms of human activities, both sites are affected by vehicle emissions, while the urban site is more affected by industrial coal combustion and the seaside site is more affected by port machinery/ship fuel emissions. Regarding marine sources, the seaside site experiences stronger impacts from sea salt and marine biological activities. In this study, we aim to highlight the common characteristics of CHOX compounds in coastal environments and the differences observed between the two site types (urban and seaside). Detailed responses are listed below, and the main correlations are marked in blue in the "Revised manuscript with changes marked" file.

**Specific:**

Line 24. Higher than what/where? Non-coastal area?

**Response:** We rewrote the sentence.

"The weighted effective oxygen to carbon content ($O_{eff}$/C) ratios (urban 0.82, seaside

0.85) indicated the highly oxidized nature of coastal compounds." (in lines 24–25)

Line 41. What do you mean "absorbed into"? Do you mean gas uptake onto aerosol particles? I would suggest change to "... could undergo a series...".
**Response:** Thanks for your suggestion. We rewrote the sentence.
"Additionally, gaseous precursors such as $SO_2$, $NOx$, and volatile organic compounds (VOCs) could undergo a series of chemical reactions to form secondary organic aerosol (SOA)." (in lines 40–42)

Line 54-56. It seems your major findings are similar to what Siegel et al (2021) and Xin et al (2024) found. I think discussions on what these two previous findings are still lacking and what is unique in your study are missing.
**Response:** These studies conclude that CHO and CHON compounds dominate urban OA, whereas S-containing species are more abundant in marine aerosols. Our study extends these observations by providing a molecular-level characterization of OA at a complex urban coastal interface. Furthermore, we reveal changes in OA composition during high $PM_{2.5}$ episodes and investigate how aerosol sources and atmospheric processes drive molecular-level variations in coastal regions. We rewrote the sentence to highlight the unique feature of our study.
"Several studies have characterized OA in different environments and have found significant variations in its molecular composition (Chen et al., 2020; Siegel et al., 2021; Zhang et al., 2024). A recurring pattern shows that CHO and CHON compounds dominate urban OA, whereas S-containing species are more abundant in marine aerosols. Nevertheless, there has been inadequate research on the molecular characterization of OA under complex conditions, such as the urban-coastal interface. This is particularly true regarding the evolution of OA composition during high $PM_{2.5}$ episodes in such environments." (in lines 51–58)

Line 68-70. Do you mean the EESI-TOF (Lopez-Hilfiker et al., 2019)? If so, it is probably still quite new in the field for aerosol particle online analysis.

**Response:** Sorry for misunderstandings in expression. We rewrote the paragraph on mass spectrometry technique.

"A variety of advanced mass spectrometry techniques, such as two-dimensional Gas Chromatograph-Electron Ionization time-of-flight Mass Spectrometry (GC×GC-EI-ToF-MS), Electrospray Ionization-Fourier Transform Ion Cyclotron Resonance Mass Spectrometry (ESI-FT-ICR MS), Extractive Electrospray Ionization time-of-flight Mass Spectrometry (EESI-TOF MS), and Filter Inlet for Gases and Aerosols-Chemical Ionization Mass Spectrometry equipped with reagent ion iodide (FIGAERO-I-CIMS), have been widely used to characterize OA compositions due to their ultrahigh mass accuracy and resolution (An et al., 2019; Cui et al., 2024; Daellenbach et al., 2024; Lopez-Hilfiker et al. 2019). However, these methods differ in their detection characteristics, including pretreatment procedures, instrumental resolution, and sensitivity toward specific compound classes. Among these, FIGAERO-I-CIMS has proven particularly effective for detecting highly oxidized, acidic, and polar organic species (Lee et al. 2014; Lopez-Hilfiker et al. 2014; Bianchi et al. 2019; Du et al., 2022; Xin et al., 2024). Moreover, FIGAERO-I-CIMS performs direct thermal desorption of filter samples, which reduces potential sample loss or compositional changes associated with conventional pretreatment procedures." (in lines 66–82)

Additional references:

Lopez-Hilfiker, F. D., Pospisilova, V., Huang, W., Kalberer, M., Mohr, C., Stefenelli, G., Thornton, J. A., Baltensperger, U., Prevot, A. S. H., and Slowik, J. G.: An extractive electrospray ionization time-of-flight mass spectrometer (EESI-TOF) for online measurement of atmospheric aerosol particles. Atmos. Meas. Tech., 12, 4867–4886. doi:10.5194/amt-12-4867-2019, 2019.

Line 85-86. I think this is probably due to not enough maintenance of the instruments? Please consider to rephase since this seems to infer that long term measurements are not reliable (?). Of course it is a challenge to run long term measurements.

**Response:** Sorry for misunderstandings in expression. We rephrased the sentence.

"However, conducting long-term online observation poses significant challenges, particularly in maintaining instrument stability and airtightness. To date, the longest such observation reported was conducted by Daellenbach et al. (2024) in Beijing, which lasted for seven months." (in lines 83–87)

Line 111-117. It seems the urban site and seaside site is very close to each other, only 18km away. How can you be sure that the urban site don't have any coastal influences at all from the transport, and vice versa? It seems to be the case as you showed in Line 274-276 that more than half of the CHO and CHON molecules at both sites shared same molecular formula accounting for 86-94% of total signals. Also in Figure S4 their spectra look very similar. Maybe digging into these molecule characteristics (e.g. diurnals, mass spectra difference, or PMF) can help on finding out more site-specific differences.

**Response:** Thanks for your suggestion. As noted previously, we have provided a further explanation regarding the sampling sites in the site description section. This study was conducted at two sampling sites (an urban site and a seaside site) in a coastal city. Both sites are influenced by anthropogenic and marine sources, but to varying degrees. In this study, we aim to highlight both the common characteristics of CHOX compounds in coastal environments and the differences observed between the two site types (urban and seaside).

It's pity that the daily filter sample cannot perform diurnal variation analysis. According to your suggestion, we have conducted a comparative analysis of mass spectrometry characteristics, as shown in Figure S9. A more detailed discussion of site-specific differences has been added in lines 318–328.

[Figure]

**Fig. S9 Mass spectrum of CHOX species signal proportion difference between urban and coastal sites.**

Line 225. Are they "significantly" higher? Did you do t-test on that (like in Line 230)?

**Response:** We conducted a paired-sample t-test on the data, and the results showed that the data from the two sites for CO, NO$x$, and SO$_2$ exhibited significant differences at the 0.001 level. We added "p<0.001, t-test" in the sentence. (in line 232)

Line 309-313. More discussions on site-specific differences would be beneficial.

**Response:** Thanks for your suggestion. In addition to the aforementioned comparison of the mass spectrometric characteristics between the two sites, we have incorporated further discussion on site-specific differences in the signal intensity distribution of CHO and CHON compounds as a function of C and O numbers (as shown in Figure S8). The detailed discussion is outlined below:

"As shown in Figures S8 and S9, the proportion of O$_6$-CHO species increased significantly with carbon number at the urban site, reaching up to 41% and 61% at C$_{14–15}$. These likely correspond to molecular formulas such as C$_{14}$H$_{16}$O$_6$ and C$_{15}$H$_{18}$O$_6$, which are oxidation products of sesquiterpenes. Under the combined influence of anthropogenic and biologic emissions, OA compounds at both sites included products from photochemical oxidation of aromatic VOCs and oxidation of biological precursors such as isoprene and monoterpenes. However, composition differed between sites, with the urban site showing higher signal intensity of species including C$_3$H$_4$O$_4$, C$_{5–6}$H$_4$O$_5$, C$_{8–9}$H$_{12}$O$_4$, and C$_6$H$_{11}$NO$_6$, while the seaside site exhibited stronger signal intensity of species such as C$_{9–10}$H$_{16}$O$_3$, C$_4$H$_{4/6}$O$_5$, C$_5$H$_8$O$_4$, and C$_9$H$_{15}$NO$_5$." (in

lines 318–328)

[Figure]

**Fig. S8 Signal intensity proportion of CHO and CHON categorized by the number of carbon atoms at the urban (a, c) and seaside (b, d) sites.**

Line 335-336. High saturation can also come from saturated fatty acids from marine source.

**Response:** We rephrased the sentence.

"The highly unsaturated CHOX compounds are likely derived from anthropogenic precursors such as aromatic VOCs and PAHs, whereas the more saturated components primarily originated from biogenic terpene compounds and saturated fatty acids from marine sources (Du et al., 2024; Chan et al., 2011; Nguyen et al., 2012; Noziere et al., 2010)."(in lines 348–352)

Line 368-370. Would be nice to add more discussions on e.g. the potential mechanisms and/or sources for these CHON compounds.

**Response:** Thanks for your suggestion. We have added further discussions on the potential mechanisms and sources for the corresponding CHON compounds.

"These compounds are likely formed through multi-generational oxidation of aromatic or biogenic VOCs, ultimately yielding products containing highly oxidized functional groups, such as -COOH and -ONO$_2$." (in lines 385–387)

Line 370-375. What's the few dominant S-containing species? Did you detect any e.g. fatty acids which FIGAERO-CIMS should be able to?

**Response:** The dominant species of S-containing compounds are Aliphatic-like OSs with low carbon content, such as $C_nH_{2n/2n+2}O_{4-5}S$ (n=2–4), as we discussed in "the classification of CHOX compounds" section. We rephrased the sentence to express it more clearly.

"Furthermore, S-containing compounds in seaside OA contained more Aliphatic-like OSs species with low carbon numbers, such as $C_nH_{2n/2n+2}O_{4-5}S$ (n=2–4), and Reduced-sulfur species (O<4) (to be discussed later), which also contributes to their relatively low weighted-OSc values." (in lines 390–393)

Line 386-389. Does it mean in both the urban site and seaside site aromatic compounds are the dominating species? If so, how could this seaside site still be classified as a seaside site? Also, in this case it's not very accurate for the conclusions in Line 404-406 where it says urban site is dominated by aromatic species while seaside OA aliphatic compounds.

**Response:** Aromatic compounds were indeed the dominant species at both the urban and seaside sites. The two sampling sites in this study are influenced by anthropogenic and marine sources, but the extent of their impacts differs. The seaside site, located closer to the coastline (2.5 km away), is subject to stronger influences from port machinery/ship fuel emissions, higher concentrations of sea salt aerosols, and marine biological activities. We have provided further details regarding these site characteristics in the site description section.

In addition, the section detailing the proportional contributions of different OA compounds has been rephrased for better clarity and accuracy.

"The categorical distribution of CHOX compounds showed similarities between the two sites, with Aromatic-like CHOX species contributing significantly to the signal at both sites (> 50%). Nevertheless, their relative abundance was markedly higher at the urban site, while the seaside site exhibited a greater proportion of Aliphatic-like compounds (42.0% vs. 39.2%)." (in lines 407–411)

"This finding demonstrates a consistent spatial pattern that urban OA is more strongly influenced by anthropogenic emissions and is dominated by aromatic species,

whereas marine-influenced OA exhibits relatively higher proportions of aged, aliphatic compounds." (in lines 423–427)

Line 400. You may change "conductive" to "vulnerable".

**Response:** We pointed out that the seaside atmosphere is more conducive to the formation of highly oxidized organic compounds.

Line 418-420. Please add some literature to support this. Because I doubt these compounds are from vehicle or combustion emissions. This would be mean both sites are dominantly affected by vehicle or combustion emissions. One the one hand, the seaside site "should" by the classification be less affected by the traffic, but if they are indeed from traffic and also dominating the signals, it means the site selection is not representative for seaside. On the other hand, $C_6H_5NO_3$ and $C_7H_7NO_3$ are well known biomass burning markers, nitrophenols, while $C_8H_{11}NO_7$ may be related to monoterpene oxidations.

**Response:** Thank you for your correction. The seaside site is indeed influenced by anthropogenic emissions, but primarily from port machinery/ship fuel emissions, and is less affected by vehicle emissions compared to the urban site. We have clarified the characteristics of both sampling locations in the revised manuscript. Although $C_6H_5NO_3$ and $C_7H_7NO_3$ are well-known biomass burning markers, given the limited biomass burning activities in the study area, we primarily attribute their presence to coal combustion (Lu et al., 2019a), industrial emissions (Lu et al., 2021) and vehicle exhaust(Lu et al., 2019b). Research also reveals that $C_7H_6O$, produced via styrene ($C_8H_8$) oxidation, is a critical precursor for $C_6H_5NO_3$, while xylene ($C_8H_{10}$) is an essential precursor for $C_7H_7NO_3$ (Xia et al., 2023). Styrene and xylene are commonly used in industrial production and are emitted during gasoline cracking. Additionally, $C_8H_{11}NO_7$ has been identified as a dominant species in laboratory SOA from the reaction of limonene with $NO_3$ radicals (Faxon et al., 2018). In the original manuscript, we mistakenly classified it as a homologue of $C_nH_{2n-7}O_xN$. Based on the above, we have rephrased the relevant sentences and added supporting literature to

strengthen our explanation.

"The signal profiles of NACs were dominated by $C_nH_{2n-7}O_xN$ homologues (e.g., $C_6H_5NO_3$ and $C_7H_7NO_3$) at both sites. Their prevalent presence is primarily attributable to formation pathways initiated by the oxidation of VOCs under anthropogenic influence, coupled with elevated NO$x$ levels (Wang et al., 2019; Xia et al., 2023; Xie et al., 2017). While biomass burning emissions were relatively limited in the study area, NACs and their aromatic VOC precursors likely originated from other combustion sources, such as coal combustion, traffic emissions, and industrial activities (Lu et al., 2019a; Lu et al., 2019b; Lu et al., 2021). Notably, NACs with DBE $\geq$ 6 exhibited higher signal intensities at the urban site, suggesting a stronger contribution from these combustion emissions." (in lines 440–450)

Additional references:

Faxon, C., Hammes, J., Le Breton, M., Pathak, R. K., and Hallquist, M.: Characterization of organic nitrate constituents of secondary organic aerosol (SOA) from nitrate-radical-initiated oxidation of limonene using high-resolution chemical ionization mass spectrometry, Atmos. Chem. Phys., 18, 5467–5481, doi:10.5194/acp-18-5467-2018, 2018.

Lu, C. Y., Wang, X. F., Li, R., Gu, R. R., Zhang, Y. X., Li, W. J., Gao, R., Chen, B., Xue, L. K., and Wang, W. X.: Emissions of fine particulate nitrated phenols from residential coal combustion in China, Atmos. Environ., 203, 10–17, doi:10.1016/j.atmosenv.2019.01.047, 2019a.

Lu, C. Y., Wang, X. F., Dong, S. W., Zhang, J., Li, J., Zhao, Y. N., Liang, Y. H., Xue, L. K., Xie, H. J., Zhang, Q. Z., and Wang, W. X.: Emissions of fine particulate nitrated phenols from various on-road vehicles in China, Environ. Res., 179, 108709, doi:10.1016/j.envres.2019.108709, 2019b.

Lu, C., Wang, X., Zhang, J., Liu, Z., Liang, Y., Dong, S., Li, M., Chen, J., Chen, H., Xie, H., Xue, L., and Wang, W.: Substantial emissions of nitrated aromatic compounds in the particle and gas phases in the waste gases from eight industries, Environ. Pollut., 283, 117132, doi:10.1016/j.envpol.2021.117132, 2021.

Wang, Y., Hu, M., Wang, Y., Zheng, J., Shang, D., Yang, Y., Liu, Y., Li, X., Tang, R., Zhu, W., Du, Z., Wu, Y., Guo, S., Wu, Z., Lou, S., Hallquist, M., and Yu, J. Z.: The formation of nitro-aromatic compounds under high NO$x$ and anthropogenic VOC conditions in urban Beijing, China, Atmos. Chem. Phys., 19, 7649-7665. doi:10.5194/acp-19-7649-2019, 2019.

Xia, M., Chen, X., Ma, W., Guo, Y., Yin, R., Zhan, J., Zhang, Y., Wang, Z., Zheng, F., Xie, J., Wang, Y., Hua, C., Liu, Y., Yan, C., and Kulmala, M.: Observations and Modeling of Gaseous

Nitrated Phenols in Urban Beijing: Insights From Seasonal Comparison and Budget Analysis, J. Geophys. Res.-Atmos., 128, e2023JD039551, doi:10.1029/2023JD039551, 2023.

Xie, M., Chen, X., Hays, M. D., Lewandowski, M., Offenberg, J., Kleindienst, T. E., and Holder, A. L.: Light absorption of secondary organic aerosol: composition and contribution of nitroaromatic compounds, Environ. Sci. Technol., 51, 11607-11616, doi:10.1021/acs.est.7b03263, 2017.

Line 431-433. Would be nice to add more discussions e.g. on the potential mechanisms and/or sources for these S-containing molecules.

**Response:** According to your suggestion, we have added further discussions on the potential mechanisms and sources for the corresponding S-containing molecules.

"The elevated abundances of Aliphatic-like OSs and Reduced-sulfur compounds in seaside OA were likely attributed to the oxidation of biogenic reduced sulfur gases, particularly dimethyl sulfide (DMS) emitted from the ocean (Shen et al., 2022; Siegel et al., 2021)." (in lines 463–466)

Additional references:

Shen, J., Scholz, W., He, X.-C., Zhou, P., Marie, G., Wang, M., Marten, R., Surdu, M., Rörup, B., Baalbaki, R., Amorim, A., Ataei, F., Bell, D. M., Bertozzi, B., Brasseur, Z., Caudillo, L., Chen, D., Chu, B., Dada, L., Duplissy, J., Finkenzeller, H., Granzin, M., Guida, R., Heinritzi, M., Hofbauer, V., Iyer, S., Kemppainen, D., Kong, W., Krechmer, J. E., Kürten, A., Lamkaddam, H., Lee, C. P., Lopez, B., Mahfouz, N. G. A., Manninen, H. E., Massabò, D., Mauldin, R. L., Mentler, B., Müller, T., Pfeifer, J., Philippov, M., Piedehierro, A. A., Roldin, P., Schobesberger, S., Simon, M., Stolzenburg, D., Tham, Y. J., Tomé, A., Umo, N. S., Wang, D., Wang, Y., Weber, S. K., Welti, A., Wollesen de Jonge, R., Wu, Y., Zauner-Wieczorek, M., Zust, F., Baltensperger, U., Curtius, J., Flagan, R. C., Hansel, A., Möhler, O., Petäjä, T., Volkamer, R., Kulmala, M., Lehtipalo, K., Rissanen, M., Kirkby, J., El-Haddad, I., Bianchi, F., Sipilä, M., Donahue, N. M., and Worsnop, D. R.: High Gas-Phase Methanesulfonic Acid Production in the OH-Initiated Oxidation of Dimethyl Sulfide at Low Temperatures, Environ. Sci. Technol., 56, 13931-13944. doi:10.1021/acs.est.2c05154, 2022.

Line 487. Would be nice to add the labels of these compounds in Figure7, similar to the compounds mentioned in Line 516. Also would be beneficial to add more discussions on their potential mechanisms and/or sources, similarly to the texts in Line 519-523.

**Response:** We have added the labels of the corresponding compounds in Figure 7a-d. According to your suggestion, we have expanded the discussions on the potential mechanisms and sources of CHON compounds.

"These organonitrates are mainly formed through the oxidation of VOCs such as alkanes, alkenes and monoterpenes by OH, $O_3$, and $NO_3$ in the presence of NO$x$ (Lee et al., 2016; Ng et al., 2017; Yan et al., 2016; Yang et al., 2025), and subsequently partition into the particle phase. Additionally, particulate organonitrates can also be generated via heterogeneous reactions of organic compounds with $NO_3$ (Nah et al., 2016)." (in lines 518–523)

Additional references:

Lee, B. H., Mohr, C., Lopez-Hilfiker, F. D., Lutz, A., Hallquist, M., Lee, L., Romer, P., Cohen, R. C., Iyer, S., Kurtén, T., Hu, W., Day, D. A., Campuzano-Jost, P., Jimenez, J. L., Xu, L., Ng, N. L., Guo, H., Weber, R. J., Wild, R. J., Brown, S. S., Koss, A., Gouw, J. de, Olson, K., Goldstein, A. H., Seco, R., Kim, S., McAvey, K., Shepson, P. B., Starn, T., Baumann, K., Edgerton, E. S., Liu, J., Shilling, J. E., Miller, D. O., Brune, W., Schobesberger, S., D'Ambro, E. L., and Thornton, J. A.: Highly functionalized organic nitrates in the southeast United States: Contribution to secondary organic aerosol and reactive nitrogen budgets. P. Natl. Acad. Sci. USA, 113, 1516-1521. doi:10.1073/PNAS.1508108113, 2016.

Nah, T., Sanchez, J., Boyd, C. M., and Ng, N. L.: Photochemical Aging of α-pinene and β-pinene Secondary Organic Aerosol formed from Nitrate Radical Oxidation. Environ. Sci. Technol., 50, 222-231. doi:10.1021/acs.est.5b04594, 2016.

Ng, N. L., Brown, S. S., Archibald, A. T., Atlas, E., Cohen, R. C., Crowley, J. N., Day, D. A., Donahue, N. M., Fry, J. L., Fuchs, H., Griffin, R. J., Guzman, M. I., Herrmann, H., Hodzic, A., Iinuma, Y., Jimenez, J. L., Kiendler-Scharr, A., Lee, B. H., Luecken, D. J., Mao, J., McLaren, R., Mutzel, A., Osthoff, H. D., Ouyang, B., Picquet-Varrault, B., Platt, U., Pye, H. O. T., Rudich, Y., Schwantes, R. H., Shiraiwa, M., Stutz, J., Thornton, J. A., Tilgner, A., Williams, B. J., and Zaveri, R. A.: Nitrate radicals and biogenic volatile organic compounds: oxidation, mechanisms, and organic aerosol. Atmos. Chem. Phys., 17, 2103-2162. doi:10.5194/acp-17-2103-2017, 2017.

Yang, Y., Huang, L., Zhao, M., Wu ,Y., Xu, Y., Li, Q., Wang, W., and Xue, L.: Multiphase reactions of organic peroxides and nitrite as a source of atmospheric organic nitrates. Nat. Commun.. 16, 5437. doi:10.1038/s41467-025-60696-3, 2025.

Same case for compounds in Line 530 and 532.

**Response:** Figure 7e-h was modified. We added the labels of the corresponding compounds.

Line 501-511. Some discussions to support explaining the observations are needed, similar to the texts in the previous and following paragraph.

**Response:** According to your suggestion, we have added further discussions on the results in the corresponding text.

"The overall increase in OSc and the high abundance of Aliphatic-like O-rich compounds can be attributed to the influence of marine air masses. Although these air masses diluted OA concentrations, they introduced aliphatic compounds and enhanced atmospheric oxidative capacity likely via abundant halogen radicals, thereby driving the observed changes." (in lines 538–542)

"The variation of CHON compositions indicates a change in the formation pathways of N-containing compounds. The notably higher RH in Case 2 would facilitate the uptake of gas-phase NACs into aerosols (Frka et al., 2016; Vidovic et al., 2018) and probably their heterogeneous reactions with $NO_3$ radicals. Concurrently, reduced UVB diminished the photolytic degradation of NACs (Peng et al., 2023). This dual effect collectively led to the observed increase in the proportion of NACs." (in lines 547–553)

Additional references:

Frka, S., Sala, M., Kroflic, A., Hus, M., Cusak, A., and Grgic, I.: Quantum chemical calculations resolved identification of methylnitrocatechols in atmospheric aerosols. Environ. Sci. Technol., 50, 5526-5535. doi:10.1021/acs.est.6b00823, 2016.

Peng, Y., Yuan, B., Yang, S., Wang, S., Yang, X., Wang, W., Li, J., Song, X., Wu, C., Qi, J., Zheng, E., Ye, C., Huang, S., Hu, W., Song, W., Wang, X., Wang, B., and Shao, M.: Photolysis frequency of nitrophenols derived from ambient measurements. Sci. Total Environ., 869, 161810. doi:10.1016/j.scitotenv.2023.161810, 2023.

Vidovic, K., Lasic Jurkovic, D., Sala, M., Kroflic, A., and Grgic, I.: Nighttime aqueous-phase formation of nitrocatechols in the atmospheric condensed phase. Environ. Sci. Technol., 52, 9722-9730. doi:10.1021/acs.est.8b01161, 2018.

**Technical:**

Line 327. Please delete "of".

**Response:** We have corrected it.

Line 502. Please delete "was".

**Response:** We have corrected it.